**Data Availability Statement:** The dataset supporting the conclusions of this article is

# The association between health literacy and quality of life of patients with type 2 diabetes mellitus: A cross-sectional study

Walid Al-Qerem[1]*, Anan Jarab[2,3,4], Alaa Hammad[1], Judith Eberhardt[5], Fawaz Alasmari[6], Safa M. Alkaee[1], Zein H. Alsabaa[7], Mahmood Al-Ibadah[8]

1 Department of Pharmacy, Faculty of Pharmacy, Al-Zaytoonah University of Jordan, Amman, Jordan, 2 College of Pharmacy, Al Ain University, Abu Dhabi, United Arab Emirates, 3 AAU Health and Biomedical Research Center, Al Ain University, Abu Dhabi, United Arab Emirates, 4 Department of Clinical Pharmacy, Faculty of Pharmacy, Jordan University of Science and Technology, Irbid, Jordan, 5 Department of Psychology, School of Social Sciences, Humanities and Law, Teesside University, Middlesbrough, United Kingdom, 6 Department of Pharmacology and Toxicology, College of Pharmacy, King Saud University, Riyadh, Saudi Arabia, 7 Department of Pharmacy, Faculty of Pharmacy, Petra University, Amman, Jordan, 8 Department of Medical Laboratory Techniques, College of Medical Technology, Al-Farahidi University, Baghdad, Iraq

* waleed.qirim@zuj.edu.jo

## Abstract

### Background

Health literacy-driven interventions in patients with type 2 diabetes have been found to play an important role in achieving glycemic control and enhancing diabetic self-management outcomes. The present study aimed to examine the role of health literacy as a predictor of quality of life among diabetic patients in Jordan.

### Methods

This cross-sectional study enrolled 400 Type 2 diabetic patients visiting the endocrinology department at the outpatient clinic at Al Basheer Hospital in Amman, Jordan. The data were collected between 1st of August and 28th of December 2023, using the validated Jordanian Diabetic Health Literacy Questionnaire and the EuroQol-5D tool. A quantile regression analysis was conducted to explore the factors associated with health–related quality of life among the study participants.

### Result

This study included 68.8% females, with a median age of 58 (50–64) years. The median EQ5-D index score was 0.66 (0.41–0.78). Findings from regression analysis indicated as patients' age increased, their quality of life scores significantly decreased (-0.004, 95%CI (-0.006, -0.001), p = 0.002). Additionally, higher JDHLQ scores were significantly associated with higher EQ5-D scores (0.012, 95% CI (0.006–0.018), p<0.001). Moreover, patients with only an elementary education had significantly lower EQ5-D scores compared to those with a postgraduate education (-0.106, 95%CI (-0.190, -0.023), p = 0.013).

available in the Zenodo repository https://doi.org/10.5281/zenodo.13133336.

**Funding:** The current project was funded by Al-Zaytoonah University of Jordan, Ref no:1/4/2022-2023. The funders had no role in study design, data collection and analysis, decision to publish, or preparation of the manuscript.

**Competing interests:** The authors have declared that no competing interests exist.

## Conclusion

The findings of the present study emphasize the importance of including health literacy assessments and interventions in the diabetes care plans of patients in Jordan.

## Introduction

Diabetes Mellitus (DM) is a cluster of metabolic diseases characterized by elevated blood glucose levels [1]. DM is caused by abnormalities in insulin action, insulin secretion, or both. The prolonged elevation of blood glucose levels in DM is associated with various complications including failure and dysfunction of different organs, such as nerves, kidneys, blood vessels, heart, and eyes [1].

According to data from the International Diabetes Federation Diabetes Atlas, the prevalence of diabetes was found to be 537 million people worldwide in 2021 [2]. The prevalence is increasing and is projected to reach 578 million by 2030 and approximately over 700 million DM patients by 2045. Prevalence rates are much higher in high-income countries compared to low-income countries [2]. In 2021, the global prevalence of DM in urban areas (12.1%) was found to be higher than in rural regions (8.3%) [3]. The prevalence of DM in Jordan is considered one of the highest globally. For example, in 2017 the prevalence was 23.7% in 2017 [3]. Furthermore, the incidence is increasing, the prevalence of DM in 1994 among men in Jordan aged 25 years or older was 14.2% and increased to 18.3% in 2004, and reached 26.8% and 32.4% in 2009 and 2017, respectively [4].

Health literacy (HL) is defined by the World Health Organization (WHO) as "the cognitive and social skills which determine the motivation and ability of individuals to gain access to, understand, and use information in ways that promote and maintain good health" [5]. HL has become increasingly important for economic, social, and health development [6]. Diabetes-related HL is the degree to which DM patients have the necessary abilities and skills to seek, analyze, understand, enumerate, and communicate DM-related information in their daily lives, clinics, and other healthcare settings [7]. HL-driven interventions in DM patients have been found to play an important role in achieving glycemic control and enhancing DM self-management outcomes [8]. Knowledge of self-care activities and how to seek and access health-related information is crucial for DM patients, as health systems are becoming increasingly complex [9]. HL has been shown to improve DM patients' health outcomes by enabling them to engage in beneficial health-related activities and perform appropriate self-care practices [10].

Quality of life (QOL) is a significant health outcome; it represents the ultimate aim of all treatments and health-related interventions [11]. QOL is a cornerstone of evaluating healthcare practice, modern medicines, and other health-related interventions [12]. Health-related QOL is a measure of an individual's perceived mental, physical, and social well-being [13]. DM patients have lower QOL than individuals with no chronic diseases, yet they tend to experience better QOL compared to patients with most other chronic illnesses [11, 14]. DM patients have lower QOL than individuals with no chronic diseases, yet they tend to experience better QOL compared to patients with most other chronic illnesses [11, 14]. According to a cross-sectional study conducted in the Al-Ahsa region of Saudi Arabia, the main problems that negatively impact the QOL of diabetic patients are depression/anxiety, mobility problems, and pain/discomfort [15]. Furthermore, a cross-sectional study carried out in northern Thailand reported that most DM patients (49.4%) had poor to moderate QOL [16].

A recent cross-sectional study on type 2 diabetes patients in Iran found that improvement in HL were associated with better QOL [17]. To date, there are no published studies evaluating

the impact of health literacy on QOL among diabetic Jordan patients. Understanding this relationship is important in addressing QOL in this patient group. Therefore, the present study aimed, and for the first time, to examine the role of HL as a predictor of QOL among patients with type 2 diabetes in Jordan.

## Materials and methods

This study involved 400 patients with type 2 diabetes attending the endocrinology department at the outpatient clinic of Al Basheer Hospital, one of the largest public hospitals in Jordan, located in East Amman and serving a significant number of patients. The data were collected from the hospital between 1st of August and 28th of December 2023. The inclusion criteria included patients diagnosed with type 2 diabetes for at least one year, being 18 years or older, literate, with the ability to read and write, since the tools used in this study are self-administered and agreeing to participate in the study by providing written informed consent. The files of the patients scheduled for follow-up appointments the next day were reviewed, and only those meeting the inclusion criteria were considered. The researchers (S.A and Z.A) briefly stated the study objectives, confidentiality of the collected information, and the participant's right to withdraw from the study at any time. Additionally, patients were informed that self-completing the questionnaire would take 10 minutes approximately. Each participant also completed an informed consent form. The study adhered to the Declaration of Helsinki's ethical guidelines. Ethical approval was obtained from the Al-Zaytoonah University of Jordan (Ref#1/4/2022–2023).

### Data collection and study instruments

The Jordanian Diabetic Health Literacy Questionnaire (JDHLQ) was adopted for this study [18], which is a validated tool used to evaluate diabetic patients' health literacy in the Arabic-speaking population. It has two sections in addition to the sociodemographic data collection sheet. Upon collecting patients' sociodemographic information, including age, gender, educational level, monthly income, and marital status, the first section is composed of five items focusing on the informative domain of health literacy, evaluating patients' ability to assess, understand, and use information about type 2 diabetes. The second section consists of items assessing communicative aspects of health literacy and patients' ability to effectively communicate about their disease, including their ability to explain the rationale for a diabetic diet, explain his/her condition to healthcare professionals, and ask them questions about type 2 doabetes. These sections collectively consist of 8 questions on a four-point Likert scale with the maximum achievable score being 32. A higher total score on this scale represents a better ability in DM-related health literacy. Data on the patient's medications and HbA1c values on the same day of the visit were collected from the patient's files.

Additionally, the EuroQol-5D (EQ-5D) [19–21], is a validated tool was used to assess QOL in Jordan [22]. This is composed of five items assessing five dimensions including usual activities, self-care, mobility, anxiety/depression, and pain/discomfort. Each dimension has three levels of response or perceived problem (Level 1: no problems, Level 2: some problems, Level 3: extreme problems/inability to perform). Each unique health state is scored on a numerical scale from -0.594 to 1. A score of one represents a perfect health state while a score of zero and lower represents death and "worse than death" (WTD), respectively [23].

### Sample size calculation

In order to calculate the minimum sample size required to produce a regression model with adequate statistical power, the 50 + 8P equation [24] was adopted, where P represents the

number of predictors. The study examined the association of 11 variables with patients' EQ-5D scores. Therefore, the minimum required sample size was 138 patients.

## Statistical analysis

Data analysis was performed using the Statistical Package for the Social Sciences (SPSS), version 26.Continuous variables were presented as medians and 25–75 percentiles, while categorical variables were presented as frequencies and percentages. The internal consistency of the EQ5-D and the informative and communitive domains of the JDHLQ were evaluated by computing Cronbach's alphas. The normality of EQ5-D index scores was assessed using Q-Q plots. Since the data was not normally distributed, nonparametric tests were conducted, along with a quantile regression analysis to examine the association between EQ5-D index scores and various variables, including gender, age, monthly income, marital status, education levels, insurance status, HbA1c, medications (Insulin, Metformin, and DPP-4 inhibitors), and JDHLQ score. Multicollinearity between the different predictors were evaluated by computing VIF values and all the values were less than 3. The $R^2$ value was measured to assess the fitness of the produced model. The significance level was set at a threshold of $p < 0.05$.

## Results

The present study enrolled 400 patients with type 2 diabetes (68.8%female). Table 1 shows the demographic characteristics of the participants. The median age was 58 (50–64) years, with the majority being married (89.2%). A significant number had only elementary education (42.5%), and most had health insurance (79.0%). Furthermore, 81.2% of patients earned less than 500 Jordanian dinars JD per month. Metformin was the most frequently used medication (86.7%), followed by Insulin (37.7%), while Thiazolidinediones (TZDs) were the least used (1.8%).

**Table 1. Sociodemographic characteristics of diabetic patients.**

| | | Median (percentile 25–75) | Count (%) |
|---|---|---|---|
| **Age** | | 58(50–64) | |
| **HbA1c** | | 8.00 (6.80–10.00) | |
| **Sex** | Female | | 275 (68.8%) |
| | Male | | 125 (31.3%) |
| **Education** | Elementary | | 169 (42.5%) |
| | High school | | 142 (35.7%) |
| | College/university degree | | 87 (21.9%) |
| **Marital status** | Single | | 43 (10.8%) |
| | Married | | 355 (89.2%) |
| **Monthly income** | less than 500 JD | | 323 (81.2%) |
| | 500 JD or more | | 75 (18.8%) |
| **Do you have health Insurance?** | No | | 84 (21.0%) |
| | Yes | | 316 (79.0%) |
| **Medications** | Insulin | | 150 (37.7%) |
| | Metformin | | 345 (86.7%) |
| | DPP-4 inhibitors | | 59 (14.8%) |
| | GLP-1-and dual GLP-1 GIP receptor agonists | | 15 (3.8%) |
| | SGLT2-Inhibitors | | 12 (3%) |
| | Sulfonylureas | | 38 (9.5%) |
| | Thiazolidinediones (TZDs) | | 7 (1.8%) |

JD: Jordanian dinar, 1 JD is equivalent to $1.4

**Table 2. Frequency of responses to diabetes-related information and diabetes-related communication items.**

| Item | Frequency (%) | | | |
|---|---|---|---|---|
| | 1 | 2 | 3 | 4 |
| Informative domain | | | | |
| Reading and understanding educational materials and booklets | 44 (11.00%) | 93 (23.30%) | 198 (49.50%) | 65 (16.30%) |
| Understand the written information I receive from my healthcare provider | 41 (10.30%) | 97 (24.30%) | 188 (47.00%) | 74 (18.50%) |
| Understand the information on diabetes management I obtain from the healthcare provider | 42 (10.50%) | 119 (29.80%) | 174 (43.50%) | 65 (16.30%) |
| Evaluate the accuracy of diabetes-related information I obtain | 73 (18.30%) | 151 (37.80%) | 122 (30.50%) | 54 (13.50%) |
| Understand the information I search for on diabetes | 43 (10.80%) | 108 (27.10%) | 175 (43.80%) | 74 (18.50%) |
| Communicative domain | | | | |
| Explain why my diabetic diet is important | 63 (15.80%) | 155 (38.30%) | 133 (33.30%) | 49 (12.30%) |
| Explaining my diabetes condition to a healthcare provider | 15 (3.80%) | 80 (20.00%) | 174(43.50%) | 131 (32.80%) |
| Ask health professionals a question | 15 (3.80%) | 66 (16.30%) | 183 (45.80%) | 136 (34.30%) |
| **JDHLQ score** | **22(18–25)** | | | |

Table 2 presents the frequency of responses to diabetes-related information and diabetes-related communication items. The ability with the highest score was for item "Understand the written information I receive from my healthcare provider", as (47.0) scored their ability as 3, and (18.50%) scored their ability at 4, and the least ability was reported for the item "Evaluate the accuracy of diabetes-related information I obtain", as only 13.5% gave themselves a rating of 4. The median for the JDHLQ score was 22 (18–25) out of a maximum possible score of 32. The Cronbach's alphas for the diabetes-related Informative and Communicative domains were 0.83 and 0.81 respectively, indicating high internal consistency.

Patients' responses to the EQ5-D items are displayed in Table 3. Regarding the mobility dimension, more than half of the patients answered, "I have some problems in walking about" (58.8%). Most had no problems with self-care (66%). However, the highest percentage of patients had some problems with performing their usual activities (48.3%), and most had moderate pain or discomfort (47.5%). Moreover, most of the patients were moderately anxious or

**Table 3. Patients' responses to EQ5-D items.**

| | | Median (percentile 25–75 | Count |
|---|---|---|---|
| Mobility | I have no problems in walking about | | 155 (38.8%) |
| | I have some problems in walking about | | 235 (58.8%) |
| | I am confined to bed | | 10 (2.5%) |
| Self-Care | I have no problems with self-care | | 264 (66%) |
| | I have some problems washing or dressing myself | | 125 (31.3%) |
| | I am unable to wash or dress myself | | 11 (2.8%) |
| Usual Activities | I have no problems with performing my usual activities | | 190 (47.5%) |
| | I have some problems with performing my usual activities | | 193 (48.3%) |
| | I am unable to perform my usual activities | | 17 (4.3%) |
| Pain/Discomfort | I have no pain or discomfort | | 77 (19.3%) |
| | I have moderate pain or discomfort | | 190 (47.5%) |
| | I have extreme pain or discomfort | | 133 (33.3%) |
| Anxiety/Depression | I am not anxious or depressed | | 135 (33.8%) |
| | I am moderately anxious or depressed | | 214 (53.5%) |
| | I am extremely anxious or depressed | | 51 (12.8%) |
| EQ5-D index score | | 0.66 (0.41–0.78) | |

Table 4. Quantile regression analysis of variables influencing the quality of life in patients with type 2 diabetes.

| Parameter | | Coefficient | Sig. | 95% Confidence Interval | |
|---|---|---|---|---|---|
| | | | | Lower Bound | Upper Bound |
| (Intercept) | | 0.639 | <0.001 | .388 | .890 |
| Age | | -0.004 | 0.002 | -0.006 | -0.001 |
| HBA1C | | <0.001 | 0.365 | -0.001 | <0.001 |
| JDHLQ | | 0.012 | <0.001 | 0.006 | 0.018 |
| DPP-4-inhibitors | No | 0.067 | 0.102 | -0.013 | 0.148 |
| | Yes | 0[c] | . | . | . |
| Insulin | No | 0.037 | 0.234 | -0.024 | 0.099 |
| | Yes | 0[c] | . | . | . |
| Metformin | No | 0.021 | 0.652 | -0.071 | 0.114 |
| | Yes | 0[c] | . | . | . |
| Educational level | Elementary school | -0.106 | 0.013 | -0.190 | -0.023 |
| | High school | -0.070 | 0.088 | -0.150 | 0.010 |
| | Postgraduate education | 0[c] | . | . | . |
| Gender | Female | -0.061 | 0.065 | -0.126 | 0.004 |
| | Male | 0[c] | . | . | . |
| Insurance status | No | 0.009 | 0.797 | -0.063 | 0.081 |
| | Yes | 0[c] | . | . | . |
| Income status | 500JOD or less | 0.003 | 0.936 | -0.075 | 0.081 |
| | More than 500jOD | 0[c] | . | . | . |
| Marital status | Single | 0.010 | 0.843 | -0.087 | 0.107 |
| | Married | 0[c] | . | . | . |

depressed (53.5%). The median EQ5-D index score was 0.66 (0.41–0.78). Cronbach's alpha of the EQ5-D was 0.8.

Findings from quantile regression (Table 4) revealed that higher JDHLQ scores were significantly associated with higher EQ5-D scores (0.012, 95% CI (0.006–0.018), p<0.001). Conversely, as patients' age increased, their QOL scores significantly decreased (-0.004, 95%CI (-0.006, -0.001), p = 0.002). Additionally, patients with only an elementary education had significantly lower EQ5-D scores compared to those who had postgraduate education (-0.106, 95%CI (-0.190, -0.023), p = 0.013). The $R^2$ value was 0.24, indicating that 24% of the variance in EQ-5D scores was explained by the model.

## Discussion

Type 2 diabetes is a chronic disease that can have a serious negative impact on patients' social, emotional, and physical health. Understanding diabetic patients' QOL offers important insights into how their condition impacts their day-to-day functioning, mental health, and other aspects of their lives [25]. Consequently, healthcare professionals can better meet the unique needs of diabetic patients by customizing interventions and support services, which will ultimately enhance their overall quality of life. Furthermore, evaluating QOL can point out areas of concern or areas that require more support, which can help inform healthcare policies and interventions meant to improve the well-being of people with type 2 diabetes.

The International Diabetes Federation (IDF) identifies the Middle East as a key region for diabetes prevalence. While standard treatments exist, preventive measures must be tailored to local cultures. Despite shared language and religion, Middle Eastern countries exhibit significant cultural, economic, and healthcare diversity. Additionally, issues like war, forced

migration, climate change, and political instability complicate healthcare delivery [26]. Urbanization, socioeconomic development, sedentary lifestyles, and high consumption of fats and sugary foods have all contributed to increasing obesity and diabetes rates, presenting significant challenges for the region [27]. Growing demands for individuals to take greater responsibility for their health have underscored the need for adequate health education. Low health literacy is seen as a significant obstacle to enhancing health outcomes. Research has consistently shown that individuals with low health literacy tend to have inadequate diabetes knowledge, engage in less effective self-management, experience poor blood glucose control, and incur higher healthcare costs [28, 29]. This issue is particularly important in the Middle East, where the rising prevalence of diabetes has become a critical concern. In a study of 256 patients with type 2 diabetes in Saudi Arabia, 27.3% exhibited marginal health literacy, while 35.5% had inadequate health literacy [30]. Another study revealed that only 11% of individuals with type 2 diabetes attending outpatient clinics across the UAE demonstrated adequate health literacy levels [31]. Thus, it is essential to develop policies and strategies that reflect the values and practices of each society. There is increasing evidence that links diabetes to a lower quality of life, with health literacy accounting for 47.5% of the variance in health-related QoL among diabetic patients [32]. However, findings on the relationship between poor health literacy and lower health-related QoL in these patients have been inconsistent [33].

The current study is the first study to assess the role of HL as a predictor of QOL among patients with type 2 doabetes in Jordan. Moderate QOL was found among the study subjects. In the current study, the median EQ-5D index score was 0.66 (0.41–0.78). Higher EQ-5D scores have been reported among diabetic patients in previous studies conducted in Jordan [34], Iran [35, 36], India [37], Nigeria [38], China [39], Ethiopia [40], and Korea [41]. The poor QOL found among type 2 patients in the present study highlights the importance of exploring the factors associated with reduced QOL in this patient group.

The current study found a significant relationship between older age and poor QOL. Similar results have been reported in earlier studies [42–46]. Older diabetic patients likely have a longer history of the disease, which increases their disease burden and their risk of complications, which can significantly lower their QOL. Additionally, they are more likely to have multiple comorbidities, which can further impair QOL. Patients with lower educational levels had significantly lower EQ-5D scores than those with higher education levels in the current study. Several previous studies have found better QOL among diabetic patients with higher educational levels [46–49]. Higher-educated patients typically have a greater understanding of their disease, including available treatment options and the possible consequences of diabetes-related complications. As a result of their greater awareness, they may be more proactive in controlling the disease, following their treatment plans, and making lifestyle modifications that will enhance their QOL [47].

Type 2 diabetes outcomes and management are significantly influenced by HL. Earlier research has shown that HL improves type 2 diabetic patients' QOL, glycaemic control, and self-care practices [30]. In the current study, higher JDHLQ scores were significantly associated with higher EQ5-D scores. A Chinese study conducted among patients with diabetic peripheral neuropathy showed that higher HL was significantly associated with improved QOL [50]. Other studies have confirmed a positive relationship between HL and QOL among patients with type 2 diabetes in Burkina Faso [51], Saudi Arabia [30], and Malaysia [52]. This is likely the case because patients with higher HL may have a better understanding of their disease and how to manage it, resulting in more effective self-care practices, better health outcomes, and higher QOL. Higher HL may also help patients communicate more effectively with healthcare providers, allowing them to receive optimal support and treatment. The present study revealed that most participants demonstrated moderate proficiency in

understanding and communicating diabetes-related information Improving HL through targeted educational interventions could improve patients' QoL

## Limitations and future research

The findings of the current study are subject to recall and social desirability biases since part of the study results were derived from self-reported data. The participants who were interested in the study's aims were more encouraged to enrol in the study, which may cause selection bias. Additionally, the results were based solely on one hospital in Jordan. However, Al-Basheer Hospital is one of the largest public hospitals in the country and serves a substantial number of patients. The current study was limited to type 2 diabetes patients and future studies assessing health literacy among patients with type 1 diabetes are deemed necessary.

## Conclusion

As the findings of the present study show that HL had a significant positive impact on patients' QOL, it emphasizes the importance of including HL assessments and interventions in the diabetes care plans of patients in Jordan. In addition, patients with higher HL were found to have better QOL. In order to help patients better understand and manage their disease, healthcare professionals should identify those who have low HL and offer tailored education and support, thereby aiming to improve QOL and type 2 diabetes management outcomes.

## Supporting information

**S1 File. Inclusivity in global research.**
(DOCX)

## Author Contributions

**Data curation:** Zein H. Alsabaa.

**Formal analysis:** Fawaz Alasmari, Mahmood Al-Ibadah.

**Funding acquisition:** Fawaz Alasmari.

**Investigation:** Safa M. Alkaee.

**Methodology:** Anan Jarab.

**Resources:** Alaa Hammad.

**Software:** Alaa Hammad.

**Writing – original draft:** Walid Al-Qerem, Judith Eberhardt, Mahmood Al-Ibadah.

**Writing – review & editing:** Walid Al-Qerem, Judith Eberhardt.

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
