## [Decision Letter · Decision Letter 0]

15 Sep 2024

PONE-D-24-32138The association between health literacy and quality of life of patients with type 2 diabetes mellitus: a cross-sectional studyPLOS ONE

Dear Dr. Al-Qerem,

Thank you for submitting your manuscript to PLOS ONE. After careful consideration, we feel that it has merit but does not fully meet PLOS ONE’s publication criteria as it currently stands. Therefore, we invite you to submit a revised version of the manuscript that addresses the points raised during the review process.

We look forward to receiving your revised manuscript.

Kind regards,

Othman A. Alfuqaha, Ph.D.

Academic Editor

PLOS ONE

**Journal Requirements:**

The current project was funded by Al-Zaytoonah University of Jordan, Ref no:1/4/2022-2023

**Additional Editor Comments:**

I hope this message finds you well. After a thorough review of your manuscript, I have made the decision to recommend Major Revisions. I kindly ask that you carefully address all comments and suggestions provided by the reviewers.

Thank you for your continued efforts, and I look forward to receiving the revised version of your manuscript.

Best regards,

Dr. Othman Alfuqaha

Academic Editor, PLoS ONE

Reviewers' comments:

Reviewer's Responses to Questions

**Comments to the Author**

1. Is the manuscript technically sound, and do the data support the conclusions?

Reviewer #1: Yes

Reviewer #2: Yes

2. Has the statistical analysis been performed appropriately and rigorously? 

Reviewer #1: Yes

Reviewer #2: No

3. Have the authors made all data underlying the findings in their manuscript fully available?

Reviewer #1: Yes

Reviewer #2: Yes

4. Is the manuscript presented in an intelligible fashion and written in standard English?

Reviewer #1: Yes

Reviewer #2: Yes

5. Review Comments to the Author

**Reviewer #1:** The association between health literacy and quality of life of patients with type 2 diabetes mellitus: a cross-sectional study

Comments for the Authors

The study discussed an important issue regarding health literacy and quality of life. The study is well written and easy to follow. Some questions and points were raised during the review process and need to be addressed.

Abstract

Please state the place where the data collection happened.

Please specify what type of diabetes. Consider this throughout the manuscript

Please elaborate on more details including study site, data collection, instruments and data analysis

In the result section: Reword to become more clear ‘The results indicated that increased age was associated with decreased HRQOL

Introduction

Please provide the most recent statistics concerning prevalence of diabetes worldwide, and is there any statistics concerning Jordan.

Is there any site for health literacy in Jordan, the introduction lacks details describing health literacy and HRQOL in Jordan.

Before The Ref [11] Please move The HRQOL paragraph to become before the health literacy paragraph

Material and Methods

Why only Type 2 diabetes were included. I would think that Type 1 would need more education concerning insulin use, could be added as future work

Please specify who collected the data

Have you evaluated the validity and reliability of the study instruments

Regarding EQ-5D REF before ref. [22] mention that this instrument has been used in several earlier studies that evaluated HRQOL among the Jordanian population

Have you conducted a univariate analysis to determine which variables to be included in the regression analysis

Results

Why did the authors use quantile regression and what is the difference between a normal one

Regarding the JDHLQ score How this could be interpreted as high, moderate, low, need improvement, ….etc

The median EQ5-D index score was 0.66 (0.41-0.78). How this could be interpreted as low, moderate, high, …etc

Regarding EQ5-D index scores (Table 3) paragraph make your focus on the association between health literacy and HRQOL, then describe the other associations

Please mention the statistics describing the fitness if the model

Discussion

Similar studies were conducted from your lab please emphasize the novelty and importance of this study.

It would be good to compare your sample with the general population when possible

**Reviewer #2:** 1. The study is interesting. However, the methods adopted is not geared to answer the research question. It is suggested that the authors develop a clear framework for DM related HL and QOL assessment. Without basic understanding of HL levels of the patient population in the region of study, the assessment of HL-QOL will generate structural flaws in HL assessment. The following points need to addressed too.

2. It is not clear of the questionnaire used to assess HL in DM patients is validated and tested for reliability given the fact that the Arab population is not homogenous, and Jordan DM patients could be patients from Palestine, and other neighboring countries.

3. How was the questionnaire administered for illiterate patients, how were consents obtained from the same study population group?

4. Was the questionnaire categorized into different domains? And how specific is the questionnaire for DM patients HL assessment? It is advised to provide a breakup of DM direct HL assessment and non-DM related HL assessment items. A reliability and validity score for each of those groups will be beneficial.

5. How was the questionnaire administered? Was it by a study coordinator or the treating physician? What were the bias in completing the questionnaire?

6. This is multicenter-multinational study, was ethics approval obtained from the ethics committees of all the participating institutions/countries?

7. How was the sample size calculated? How many patients with DM withing the inclusion criteria are seen at the study site annually? There is a high probability that the study is underpowered.

8. The results of the study oscillate between HL and QOL. What was the HL rate among the DM patients? Is there a correlation between sociodemographic and HL, and does that impact QOL in DM patients?

9. The authors need to explore literature very well, a few landmark studies from the middle east allude to very low levels of HL among DM patients, and the challenges in DM care. Designing a health literacy model for patients with diabetes | BMC Health Services Research | Full Text (biomedcentral.com) and BMJ Open 2023 Feb 6;13(2):e069489.doi: 10.1136/bmjopen-2022-069489.

6. PLOS authors have the option to publish the peer review history of their article (what does this mean?). If published, this will include your full peer review and any attached files.

Reviewer #1: No

Reviewer #2: **Yes: **Satish C Nair

---

## [Author Response · Author response to Decision Letter 0]

26 Sep 2024

All the require amendments were performed, however, please note the Expected Publication Fee displayed was $2290.00, however, the funding organization and the corresponding author are from Jordan and we are entitled for a discount that we would like to request

---

## [Decision Letter · Decision Letter 1]

15 Oct 2024

The association between health literacy and quality of life of patients with type 2 diabetes mellitus: a cross-sectional study

PONE-D-24-32138R1

Dear Dr.

We’re pleased to inform you that your manuscript has been judged scientifically suitable for publication and will be formally accepted for publication once it meets all outstanding technical requirements.

Kind regards,

Othman A. Alfuqaha, Ph.D.

Academic Editor

PLOS ONE

Additional Editor Comments (optional):

Dear authors,

I am writing to offer my sincere congratulations on the successful completion and review of your cross-sectional study on the quality of life among participants with Type 2 diabetes mellitus. Your dedication to addressing the reviewers' comments with care and precision has not gone unnoticed, and it is clear that your hard work has greatly contributed to the strength and rigor of your research.

It is particularly commendable that this important study was conducted with funding solely from Al-Zaytoonah University.

Once again, congratulations on this significant achievement, and I wish you continued success in your future endeavors

Reviewers' comments:

Reviewer's Responses to Questions

**Comments to the Author**

1. If the authors have adequately addressed your comments raised in a previous round of review and you feel that this manuscript is now acceptable for publication, you may indicate that here to bypass the “Comments to the Author” section, enter your conflict of interest statement in the “Confidential to Editor” section, and submit your "Accept" recommendation.

Reviewer #1: All comments have been addressed

Reviewer #2: (No Response)

2. Is the manuscript technically sound, and do the data support the conclusions?

Reviewer #1: Yes

Reviewer #2: Partly

3. Has the statistical analysis been performed appropriately and rigorously? 

Reviewer #1: Yes

Reviewer #2: No

4. Have the authors made all data underlying the findings in their manuscript fully available?

Reviewer #1: Yes

Reviewer #2: No

5. Is the manuscript presented in an intelligible fashion and written in standard English?

Reviewer #1: Yes

Reviewer #2: Yes

6. Review Comments to the Author

Reviewer #1: The authors addressed all the comments in the first revision and made the ammendment needed in the text manuscript

Reviewer #2: The authors have to carefully go through the reviewer comments and revise the manuscript. Additionally, the authors also need to provide a point-by-point response to the reviewer queries on a separate note.

7. PLOS authors have the option to publish the peer review history of their article (what does this mean?). If published, this will include your full peer review and any attached files.

Reviewer #1: No

Reviewer #2: **Yes: **Satish Chandrasekhar Nair

---

## [Editor Report · Acceptance letter]

22 Oct 2024

PONE-D-24-32138R1 

PLOS ONE

Dear Dr. Al-Qerem, 

I'm pleased to inform you that your manuscript has been deemed suitable for publication in PLOS ONE. Congratulations! Your manuscript is now being handed over to our production team.

Kind regards, 

on behalf of

Dr. Othman A. Alfuqaha 

Academic Editor

PLOS ONE